# Sensing the allosteric force

Olga Bozovic [1], Brankica Jankovic [1] & Peter Hamm [1]✉

Allosteric regulation is an innate control in most metabolic and signalling cascades that enables living organisms to adapt to the changing environment by tuning the affinity and regulating the activity of target proteins. For a microscopic understanding of this process, a protein system has been designed in such a way that allosteric communication between the binding and allosteric site can be observed in both directions. To that end, an azobenzene-derived photoswitch has been linked to the $\alpha$3-helix of the PDZ3 domain, arguably the smallest allosteric protein with a clearly identifiable binding and allosteric site. Photo-induced *trans*-to-*cis* isomerisation of the photoswitch increases the binding affinity of a small peptide ligand to the protein up to 120-fold, depending on temperature. At the same time, ligand binding speeds up the thermal *cis*-to-*trans* back-isomerisation rate of the photoswitch. Based on the energetics of the four states of the system (*cis* vs *trans* and ligand-bound vs free), the concept of an allosteric force is introduced, which can be used to drive chemical reactions.

[1] Department of Chemistry, University of Zurich, Zurich, Switzerland. ✉email: peter.hamm@chem.uzh.ch

Protein allostery represents the intriguing phenomenon of communication between two distant sites of a protein, which allows for reversible modulation of protein activity. Due to its complex nature and the plethora of mechanisms proteins use to fine-tune their activity, protein allostery continues to be one of the most interesting unresolved problems in biochemistry today[1–10]. Almost all allosteric systems are large multi-domain proteins; so large that the investigation of the mechanism in atomistic detail is very difficult, not only because large proteins present an experimental challenge, but also because they are beyond what can currently be performed by molecular dynamics simulations[9,11–14].

One exception in this regard is the PDZ3 domain of the postsynaptic density-95 (PSD-95) protein[15]. As an integral part of a multi-domain protein, the PDZ3 domain plays a role in both binding the C-terminus of target proteins and influences the overall regulation of PSD-95[16,17]. In the context of the PDZ domain family, PDZ3 stands out regarding its characteristic structure. That is, besides the common and conserved secondary structure elements, PDZ3 has a C-terminal extension that forms an additional third α-helix (α3-helix), which packs against the core domain and is located at the opposite side from the binding pocket[18,19]. Further exploring its function, Petit et al.[15] showed that even though the α3-helix does not form direct contact with the peptide ligand, its deletion dramatically decreases the binding affinity for a 7-mer CRIPT-derived peptide ligand by 21-fold. In the same study, it was evidenced that the global conformation of the PDZ3 domain does not change regardless of the presence of the α3-helix. The only significant change was found to be entropic in nature and was ascribed to the changes in the dynamics of the side chains. This revealed that the α3-helix is the allosteric element and established the PDZ3 as an ideal model system to study and control allostery within a single small domain.

In this study, we designed a photocontrollable PDZ3 domain (Fig. 1) by incorporating an azobenzene-based photoswitch into the allosteric element—the α3-helix. Photocontrol is a powerful tool for altering the properties of biomolecules, and it has been exploited in many different ways in the context of peptides and proteins[20–29]. For allostery to be controlled by light in a reversible manner, the emphasis lies on both the binding and the allosteric site. We exploit different observables that allow for the examination of allosteric communication between those sites in both directions. On the one hand, we demonstrate a dramatic difference in the binding affinity as a consequence of selective perturbation of the α3-helix. In the opposite direction, allosteric communication manifested itself via modulation of the thermal cis-to-trans isomerisation kinetics of the photoswitch induced by ligand binding. Based on the latter effect, we will introduce the concept of an allosteric force.

## Results

**Designing reversibly photocontrollable allostery.** Our starting premise was to maximally perturb the α3-helix in one configuration of the photoswitch. A spacing of seven amino acid residues was previously shown to be compatible with the cis configuration, stabilising a helical structure, while the trans configuration disrupts it[20,21]. In addition, the anchoring points needed to be solvent exposed in order not to affect the global protein fold. The first anchoring point was therefore chosen to be Glu395, while the second anchoring point was Lys402. These residues were mutated to cysteines in order to facilitate the incorporation of the photoswitch element (see Fig. 1, mutated residues are shown in pink)[25,29].

As a ligand, we chose a 5-mer peptide (KETWV) due to the following reasons: An as short as possible peptide is desirable in order to avoid any direct contact between a floppy N-terminus of the ligand and the α3-helix of the PDZ3 domain. That five amino acid peptide is the shortest possible sequence, which still includes the crucial amino acids for specific binding to the PDZ3 domain (KETWV)[30]. Furthermore, the tryptophan residue at the position −1 of the peptide allows for the determination of the binding affinity for the photocontrollable PDZ3 system by fluorescence quenching. This is crucial, given that ITC would not work due to the fast thermal cis-to-trans isomerisation of the photocontrollable element (in contrast to fluorescence measurements, constant illumination of the sample is not possible in ITC). Synthesis of protein and peptide are described in "Methods".

The peptide binds to the wild-type PDZ3 domain with a reasonable binding affinity, as measured by ITC ($25 \pm 3\,\mu M$ at 21 °C) as well as by intrinsic fluorescence quenching ($24 \pm 2\,\mu M$), see SI, Fig. S2. The ITC result can be fit assuming one binding site, evidencing that it is specific binding equivalent to that shown in Fig. 1, left (which however is the X-ray structure of a slightly different peptide).

In all subsequent experiments with the photoswitchable PDZ3 domain, the trans-state has been measured as dark-adapted protein (for at least 24 h at room temperature), when it reaches essentially 100% of the trans-state. The cis-state has been measured during, or very shortly after, illumination at 370 nm. We verified by UV/Vis spectroscopy (see SI, Fig. S3) that the photo-equilibrium then reaches ≳85% cis. None of the reported numbers in the cis state have been corrected for the small residual trans-contribution.

The protein is stably folded up to ~40 °C in both states of the photoswitch, as evidenced by circular dichroism (CD) spectroscopy (see SI, Fig. S4). As known for the wild type[31,32], it exhibits two unfolding steps. However, subtle differences in the two-CD spectra indicate a small but reproducible difference in the helicity upon photoswitching, with the cis-state being the one with larger helical content, as anticipated (Fig. 2a). The difference between the CD spectra of the cis and the trans configuration resembles the characteristic response of a completely α-helical system with two minima around 210 and 225 nm (Fig. 2a, inset)[33–35]. The change in absolute ellipticity at 222 nm is ≈5%, as compared to complete unfolding of the protein induced by urea (see SI, Fig. S5). Given that the α3-helix accounts for ≈6% of the overall protein, we conclude that photoswitching significantly perturbs the conformation of the distal α3-helix.

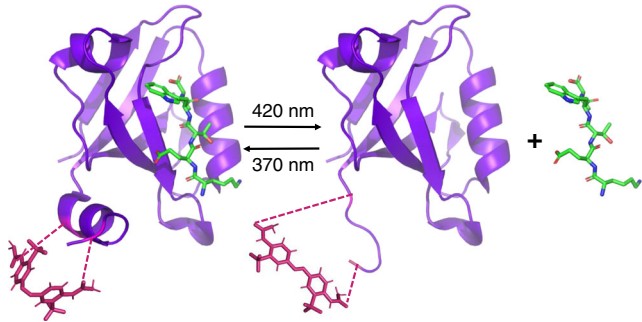

**Fig. 1 Schematic display of the photocontrollable PDZ3 system complexed with the KETWV peptide.** The left panel shows the cis configuration, which promotes the α3 helix, and the right panel the trans configuration, which disturbs it. The cysteine residues chosen as anchoring points for the photoswitch are shown in pink. The ligand-bound structure has been adapted from an X-ray structure PDB:1tp5[52], which is for a somewhat longer peptide (KKETWV) bound to PDZ3, by removing one N-terminal lysine to reveal the peptide ligand used here.

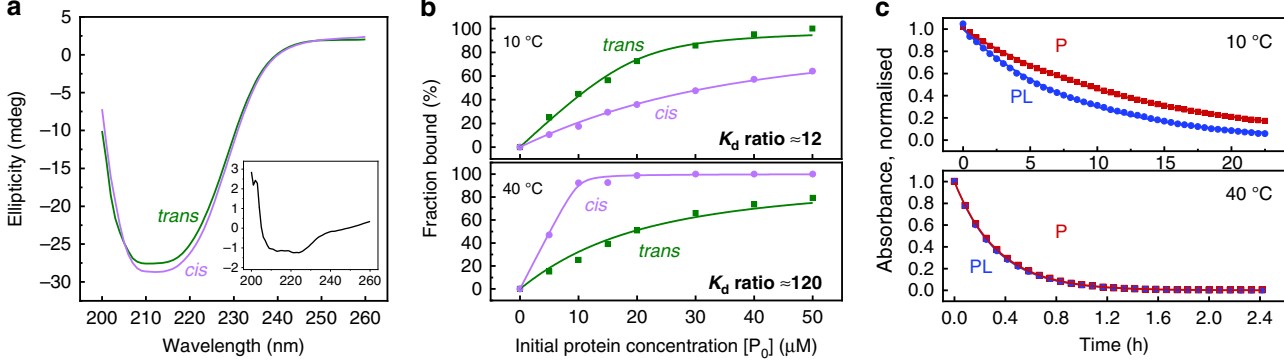

**Fig. 2 Spectroscopic observables. a** CD spectra of the photocontrollable PDZ3 domain in *trans* (green) and *cis* (violet), showing the difference in helicity between the two configurations of the photoswitch. Inlet graph shows the *trans* to *cis* CD difference spectrum, which corresponds to the CD signal of an isolated helical protein. **b** Fraction of bound ligand as a function of initial protein concentration $[P_0]$ with the initial ligand concentration $[L_0]$ kept fixed at 15 µM. Data are shown for the photoswitch in *trans* (green) and *cis* (violet) configurations and at 10 °C (top) and 40 °C (bottom). **c** Thermal *cis*-to-*trans* relaxation of the photocontrollable PDZ3 alone (red), and with saturating concentrations of the ligand (blue) at 10 °C (top) and 40 °C (bottom). Source data are provided as a Source Data file.

**Controlling allostery in both directions**. In Fig. 2b, we measured the effect of photoswitching the $\alpha$3-helix on the binding affinity for both states of the protein, using intrinsic tryptophan fluorescence quenching. The binding affinity was determined by fitting the data to a two-state binding equilibrium:

$$K_d = \frac{[P] \cdot [L]}{[PL]} = \frac{([P_0] - [PL]) \cdot ([L_0] - [PL])}{[PL]}, \qquad (1)$$

where $[P]$, $[L]$, and $[PL]$ are the concentrations of free protein, free ligand, and protein-ligand complex, respectively, after the system reached equilibrium, and $[P_0]$ and $[L_0]$ the initial concentrations of protein and ligand added to the solution. Figure 3a compiles the essence of this analysis by showing van-t'Hoff plots of the extracted binding constant $K_d$ as a function of temperature in the two states of the photoswitch. At 21 °C, the binding affinity in both the *cis* and the *trans*-states of the protein (7.0 ± 0.2 µM and 3.8 ± 0.2 µM, respectively), is larger than that for the wild-type protein (25 µM), evidencing that the protein is structurally more constrained in either case. The binding affinity greatly increases with temperature in the *cis*-state, while that of the *trans*-state decreases slightly. As a consequence, the effect inverts in the considered temperature range, with a ≈120-fold greater binding affinity of the *cis*-state at 40 °C vs a ≈12-fold stronger binding affinity in the *trans* state at 10 °C. This inversion of behaviour in a relatively small temperature range is remarkable. From the slope of the plots in Fig. 3a, the binding enthalpies can be determined, which are $\Delta H_{cis} = 130 \pm 30$ kJ/mol and $\Delta H_{trans} = -50 \pm 2$ kJ/mol, respectively. Most notably, binding is endothermic in the *cis*-state with a large entropic stabilisation.

Figure 2c shows the thermal *cis*-to-*trans* isomerisation rates with either a peptide ligand bound to the protein (using a 40-times excess of peptide to ensure 100% binding) or the free protein. To that end, we first accumulated the *cis*-state of the photoswitch by illumination at 370 nm, and then measured the thermal back-reaction via its UV/Vis spectrum. The essence of these kinetic studies is compiled in the Arrhenius plots of Fig. 3b, which reveal the activation enthalpies with $\Delta H_{PL}^{\#} = 79 \pm 1$ kJ/mol and $\Delta H_P^{\#} = 89 \pm 3$ kJ/mol, respectively. The thermal *cis*-to-*trans* isomerisation rate thus feels the presence of a ligand bound to the binding pocket of the protein (Table 1).

## Discussion

By constructing a photocontrollable PDZ3 domain, we can control allostery between binding and allosteric site in both directions. On the one hand, isomerising the photoswitch attached to the $\alpha$3-helix has a huge effect on the binding affinity, at some temperatures even larger (i.e., 120-fold at 40 °C) than cleavage of the helix (21-fold)[15]. On the other hand, ligand binding affects the rate of thermal *cis*-to-*trans* isomerisation of the azobenzene

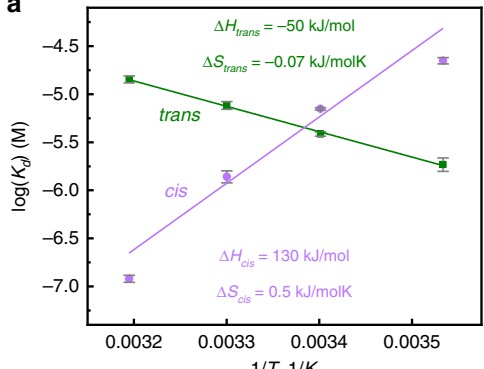

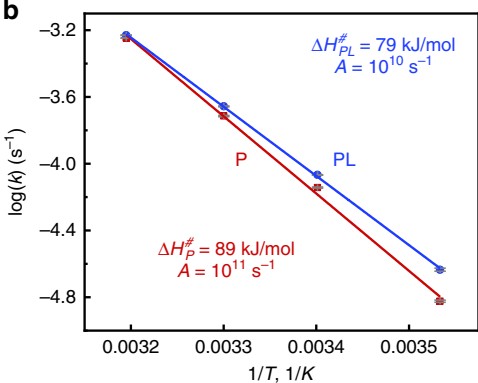

**Fig. 3 Energetics. a** Van-t'Hoff plots of the binding affinity of the peptide ligand to the photoswitchable PDZ3 domain in its *cis* (magenta) and *trans*-state (green). The binding enthalpies $\Delta H$ and entropies $\Delta S$ extracted from the linear fits are indicated. **b** Arrhenius plots of the thermal *cis*-to-*trans* isomerisation rate for the free protein ($P$, in red) or with the ligand bound ($PL$, in blue). The activation enthalpies $\Delta H^{\#}$ and the pre-exponential factors $A$ are indicated. All data are also listed in Table 1, and the corresponding titration and relaxation measurements are shown in SI as Figs. S6 and S7. Error bars, which have been estimated from the fits of the data shown in Figs. S6 and S7, indicate single standard deviation around the mean. Source data are provided as a Source Data file.

**Table 1 Binding affinities and *cis*-to-*trans* isomerisation time-constants as a function of temperature.**

|         | $K_{d,trans}$ (μM) | $K_{d,cis}$ (μM) | $\tau_P$ (h)      | $\tau_{PL}$ (h)    |
|---------|--------------------|------------------|-------------------|--------------------|
| 10 °C   | 1.8 ± 0.3          | 22 ± 2           | 12.8 ± 0.2        | 8.3 ± 0.2          |
| 21 °C   | 3.8 ± 0.2          | 7.0 ± 0.2        | 2.68 ± 0.02       | 2.25 ± 0.01        |
| 30 °C   | 7.6 ± 0.7          | 1.4 ± 0.2        | 0.99 ± 0.01       | 0.87 ± 0.01        |
| 40 °C   | 14.2 ± 1.2         | 0.12 ± 0.01      | 0.340 ± 0.003     | 0.327 ± 0.003      |

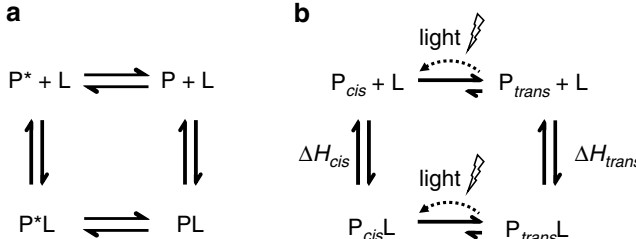

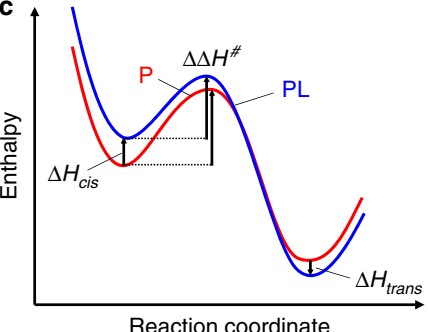

**Fig. 4 Thermodynamic cycles. a** Conventional thermodynamic cycle of the two-state model of allostery with *P* and *P\** being resting and active states of the protein, and *P* + *L* and *PL* the apo and ligand-bound states, respectively. In contrast, **b** the thermodynamic cycle of the photoswitchable system, where the *cis*-to-*trans* reaction is essentially unidirectional, but can transiently be put out of equilibrium by light, as indicated by the flash and the curved dotted arrow. **c** Energy profiles (not on scale) of the thermal *cis*-to-*trans* isomerisation of the free protein *P* (red) or the ligand bound protein *PL* (blue).

moiety (there are a few reports in literature with similar observations[36–38], which however have been discussed in a different context.) Furthermore, one may consider the *cis*-to-*trans* isomerisation a chemical reaction, as a chemical bond, the N=N double bond of the azobenzene moiety, is broken and reformed differently. Consequently, the construct is an allosteric system in its literal sense, where ligand binding on a remote site affects the rate of a chemical reaction. It is arguably the smallest known truly allosteric system; small enough to study it in atomistic detail, e.g., by NMR and IR spectroscopy or by molecular dynamics (MD) simulations.

Entropy plays a major role in ligand binding, as particularly seen for the *cis*-state, which is endothermic (Fig. 3a). Thermal *cis*-to-*trans* isomerisation, in contrast, is in essence a unimolecular isomerisation process of a small molecule, whose entropic contribution to the transition state is typically small[39–41]. In fact, the pre-exponential factors deduced from the Arrhenius plots in Fig. 3b ($1 \times 10^{11}\,\mathrm{s}^{-1}$ for *P* and $1 \times 10^{10}\,\mathrm{s}^{-1}$ for *PL*) do not differ dramatically from that predicted by the Eyring equation, $k_B T/h = 6.3 \times 10^{12}\,\mathrm{s}^{-1}$, evidencing that they reflect in essence the search rate due to thermal motion, without too much of a contribution from an activation entropy. In other words, the rate-limiting step of barrier crossing affects the central N=N group of the azobenzene moiety only very locally, as also evidenced by MD simulations of light-induced isomerisation reactions in similar azobenzene-protein systems[25,42].

Given that the isomerisation kinetics is mostly controlled by the activation enthalpy of thermal *cis*-to-*trans* isomerisation, we consider a thermodynamic cycle of enthalpy *H*. Fig. 4a shows the conventional thermodynamic cycle, which is commonly plotted in the context of the two-state model of allostery[43–45]. In this case, all reactions, ligand-binding and unbinding as well as the transition between the active (*P\**) and resting state (*P*) of the protein, are equilibrium reactions with free-energy differences in the order of $k_B T$. Hence, thermally driven transitions are possible in both directions (Fig. 4a). Bidirectional communication in allostery is the direct consequence of such a thermodynamic cycle. The important difference in the present system (Fig. 4b) is that the *cis*-to-*trans* reaction has a driving force much larger than $k_B T$ and the reaction is in essence unidirectional (indicated by the short *trans*-to-*cis* arrows in Fig. 4b). Light, however, introduces an additional knob of control that transiently puts the system out of equilibrium, as indicated by the flash and the curved dotted arrows in Fig. 4b. This knob introduces an action-reaction principle, according to which allosteric control is bidirectional (to contrast it to bidirectional allosteric communication). That is, ligand binding affects the energetics of the chemical reaction, which an allosteric system is supposed to drive. If, in contrast, one can control that chemical reaction independently, made possible here by light, it will inevitably affect ligand binding.

The driving force for thermal *cis*-to-*trans* isomerisation of the photoswitch is larger with the ligand bound, since $\Delta H_{cis} > \Delta H_{trans}$. However, the driving force per se does not necessarily determine the rate of a chemical reaction; for that the activation energy is

relevant. Figure 4c compiles the energetics of the *cis*-to-*trans* isomerisation, where $\Delta H_{cis}$ and $\Delta H_{trans}$ have been determined from the binding equilibria (Fig. 3a), and the difference in barrier heights $\Delta\Delta H^{\#} = \Delta H_{PL}^{\#} - \Delta H_{P}^{\#}$ from the thermal *cis*-to-*trans* isomerisation rates (Fig. 3b). The results suggest that the activation barrier is mostly, but not completely, *cis*-like. That is, the ligand-bound state *PL* is energetically higher by $\Delta H_{cis} = 130\,\mathrm{kJ/mol}$ as compared to the unbound state *P*. If the transition state would be completely *cis*-like, it would just parallel that energy shift. However, from the kinetics (Fig. 3b), we can deduce that $\Delta\Delta H^{\#} = \Delta H_{PL}^{\#} - \Delta H_{P}^{\#} = -10\,\mathrm{kJ/mol}$, hence the transition state follows the energy shift of the *cis*-state to only 92%. Presumably, the barrier is mostly *cis*-like since the event of *cis*-to-*trans* isomerisation is the initial process, which is fast, while the protein adapts to the change on a much slower timescale.

The spirit of this analysis is the same as that of a *Φ*-value analysis used in the context of protein folding[46], which also relates the folding free energy, determined thermodynamically, to the height of the folding barrier, measured kinetically. The *Φ*-value analysis is closely related to the linear free-energy relationship common in chemical kinetics, which is used as a means of analysing structures of transition states, and acknowledges that the activation energy of a chemical reaction is often correlated with its driving force. The linear free-energy relationship has been discussed in the context of allostery before[47]. Just like for a *Φ*-value analysis, there will be cases where ligand binding has little effect on the isomerisation kinetics of a chromophore in a protein[38], and others with a large effect[37].

The action of the photoswitch, in essence, is a change in distance between the two anchoring points by $\Delta x \approx 3\,\text{Å}$, as estimated

from MD simulations[22]. It is important to stress that this is not a distance change induced by a rigid entity, as the azobenzene moiety of the photoswitch would probably be. Rather, there are linkers with six, to a certain extent flexible single bonds on each side between the benzene rings and the protein backbone. We therefore think of the distance change as the result of a force induced by the azobenzene moiety and transmitted to the protein backbone via these linkers. Together with the difference in driving force for the *cis*-to-*trans* isomerisation, $\Delta \Delta H = \Delta H_{cis} - \Delta H_{trans} = 180$ kJ/mol, we can estimate that force, $\Delta F = \Delta \Delta H / \Delta x \approx 1$ nN. We call it an allosteric force. The force is sizeable in comparison to what is typically observed in pulling experiments on proteins (on the order of 10–100 pN), however, when comparing these numbers, one must keep in mind that force spectroscopy typically works on significantly longer length scales, 1–10 nm[48].

In the *cis*-state of the photoswitch, ligand binding is endothermic and strongly entropy driven, which is quite unusual. According to an extensive study, binding of a large variety of peptides to the PDZ3 domain is exothermic in all cases with positive or negative entropic contributions, the latter of which however never of the size observed here[30]. Apparently, the *cis*-configuration of the photoswitch compresses the α3-helix, thereby strongly restricting its configurational space. Ligand binding pushes the α3-helix apart against the force of the azobenzene moiety, which costs enthalpy but increases the overall configurational space. It is that force, which eventually speeds up the *cis*-to-*trans* isomerisation. While this certainly is a speculative explanation, we consider it a working hypothesis to be tested by MD simulations, which naturally include the calculation of forces.

In conclusion, we can not only use the azobenzene moiety to control a protein system by photoswitching, as demonstrated for numerous examples before[20–29], we can also use it to sense and quantify allosteric forces inside a protein. Ligand binding induces such an allosteric force at a remote site of the protein, where it can be used to control chemical reactions that involve conformational changes.

## Methods

**Protein and peptide preparation.** The wild-type PDZ3 gene (residues 302–403, numbering scheme of Doyle et al.[18]) from the PSD-95 and the gene containing two amino acid residues mutated into cysteines (residues 395 and 402) were cloned into a pET30a(+)vector. Plasmids (pET-30a(+)) for the wild-type PDZ3 and photoswitchable PDZ3 variant were ordered from GenScript. The genes for both proteins were inserted into NdeI-XhoI cloning site. Sequences of both genes are provided in the Supplementary Table S1. Tobacco echt virus (TEV) protease cleavage site was added between the N-terminal hexahistidine tag (His-tag) and the protein sequence. The PDZ3 domain was expressed in *E. coli* BL21(DE3) in Luria-Bertani medium using a standard protocol[25]. The protein was purified from bacterial cells using HisPrep column (GE Healthcare Life Sciences) in 50 mM TrisHCl, 200 mM NaCl and 25 mM imidazole, pH 8.5, and eluted with 50 mM TrisHCl, 200 mM NaCl and 500 mM imidazole. Prior to the linking reaction with the photo-controllable linker BSBCA, cysteine residues were reduced using 50 mM TCEP as described elsewhere[25]. The linking reaction was performed in denaturing conditions (6 M guanidinium, 50 mM TrisHCl, pH 8.5) and 50 °C under an inert atmosphere for the duration of 12 h in order to maximise the yield. The reaction mixture was subsequently desalted, and the linked protein was purified using MonoQ anion exchange column (GE Healthcare Life Sciences) in 50 mM TrisHCl, pH 8.5 with a NaCl gradient. The removal of the His– tag from the proteins was performed as described elsewhere (the required vector for pRK793, Addgene plasmid 8827, was a gift from David Waugh)[49,50]. The cleaved protein was purified using HisTrap column (GE Healthcare Life Sciences), and desalted against 20 mM NaPi, 15 mM NaCl, pH 6.8 buffer. The peptide ligand (KETWV) was synthesised using standard Fmoc-based solid-phase peptide synthesis (SPPS) on valine-preloaded Wang resin and purified as described previously[29]. The peptide was subsequently dialysed against appropriate buffer and lyophilised. The purity of all samples was confirmed with mass spectrometry (see Fig. S1), and the concentration of all samples was confirmed with amino acid analysis.

Mass spectrometry was performed by ESI-MS on a Synapt G2-Si mass spectrometer and the data were recorded with the MassLynx 4.2 Software (both Waters, UK). Prior to ESI-MS analyses, the protein samples were desalted using C4 ZipTips (Millipore, USA) and analysed in MeOH:2-PrOH:0.2% FA (30:20:50); the

peptide sample was merely diluted. The solutions were infused through a fused silica capillary (ID75 μm) at a flow rate of 1 μL/min and sprayed through a PicoTip (ID30 μm). The last was obtained from New Objective (Woburn, MA). Mass spectra were acquired in the positive-ion mode by scanning an *m/z* range from 100 to 5000 da with a scan duration of 1 s and an interscan delay of 0.1 s. The spray voltage was set to 3 kV, the cone voltage to 50 V, and source temperature 80 °C. The recorded *m/z* data of the protein samples (wild-type protein Fig. S1a and photocontrollable PDZ3 Fig. S1b)were then deconvoluted into mass spectra by applying the maximum entropy algorithm MaxEnt1 (MaxLynx) with a resolution of the output mass 0.5 Da/channel and Uniform Gaussian Damage Model at the half height of 0.7 Da. The peptide (Fig. S1c) was detected as a singly charged ion and depicted without any further processing.

**Binding affinity of the wild-type protein.** The binding affinity of the wild-type protein was measured by ITC as well as intrinsic fluorescence quenching (see Fig. S2a, b), in order to cross-validate the two methods. The former was carried on MicroCal ITC200 (software version 1.26.0, Malvern, UK) at 21 °C. The measurements were performed in duplicate in order to confirm the reproducibility of the data. The cell was loaded with 250 μL of 500 μM solution of protein, while the syringe was loaded with a 5 mM solution of the peptide. The thermograms were analysed with the standard Origin ITC software provided by the manufacturer (MicroCal ITC200 Origin 7 SR4 software version v7.0552). The ITC results can be fit assuming one binding site, evidencing that it is specific binding. The arithmetic mean value for the $K_d$ obtained from for two independent measurements is 25.2 ± 2.2 μM (see Fig. S2a). The result from intrinsic fluorescence quenching is 24.6 ± 1.2 μM (see Fig. S2b), and is in excellent agreement with the ITC result.

**Determining the photo-equilibrium.** In order to estimate the percentage of conversion to the *cis*-state upon illumination with a 370 nm continuous-wave diode laser (100 mW, CrystaLaser), we followed the same procedure as described previously[29]. After recording the UV–Vis spectrum (Shimadzu UV-2450 UV/Vis) of the *trans* and *cis* state of the protein, we calculated the ratio of absorbances at 370 nm (maximum absorption wavelength of the *trans* state) and approximated that the absorbance of the pure *cis* state would be zero at the respective wavelength. From that, the purity of the *cis*-state can be estimated to be at least 85% (see Fig. S3).

**Circular dichroism spectroscopy.** Circular dichroism spectra were collected on Jasco J-810 spectropolarimeter with a Jasco PFD-425S Peltier. For the thermal stability of both wild-type and photocontrollable PDZ3 domain ellipticity at 220 nm was recorded as a function of temperature (see Fig. S4). The thermal stability of the *cis*-state was performed as described previously[51]. As reported previously in the literature, the PDZ3 domain exhibits a three-state unfolding behaviour[31]. Data were fitted to a two-step sigmoidal function using OriginPro2018b. Values obtained for the transitions of the wild-type PDZ3 (67.7 ± 3.8 °C and 79.5 ± 0.6 °C) are in agreement with previous findings (70.4 ± 0.5 °C and 79.2 ± 1.2 °C)[32]. The values obtained for the transitions of the *trans* and *cis*-state of our photoswitchable PDZ3 domain (52.3 ± 2.5 °C, 79.2 ± 0.3 °C and 45.2 ± 0.7 °C, 86.7 ± 0.5 °C, respectively) confirm that the protein is thermally stable up to 40 °C.

For the difference in helicity upon photoswitching, the spectrum of the native protein was first recorded in the dark and then illuminated with 370 nm continuous-wave laser (CrystaLaser, power ≈ 90 mW) for the duration of three minutes, immediately after which the *cis* spectrum was collected. The procedure was repeated for the protein denatured with 8 M urea. 8 M urea solution was used rather than thermally denaturing the protein in order to make sure no residual secondary structures remained. In order to calculate the extent of perturbation upon photoswitching, the ellipticity value at 222 nm for the denatured protein was taken as 0%, and the *cis* state as 100% folded (see Fig. S5). All spectra were collected at 21 °C.

**Intrinsic tryptophan fluorescence quenching.** Fluorescence quenching of the peptide, which contains one tryptophan residue, was used to measure the binding affinity between PDZ3 and the peptide by varying the protein concentration (0–50 μM) and keeping the peptide concentration constant (15 μM). Measurements were performed using PerkinElmer fluorometer in a 1.5 mm quartz cuvette. The excitation wavelength was set to 250 nm, and emission was detected at 325 nm (both are isosbestic points of the photoswitch absorption). Samples that contained only the protein were recorded as well, but the contribution from the protein in both states (*cis* and *trans*) was negligible. *Trans* measurements were performed after incubation of the sample in the dark, while *cis* measurements were performed after 5 min pre-illumination of the sample with a 370 nm continuous wave laser and under continuous illumination during the measurement. Measurements at all temperatures were done using the same stock solution mixture of the protein and peptide. Binding curves were constructed using the value of the peptide alone as 0% bound and the plateau value (with saturating protein concentration achieved with the stronger binder) as 100% bound for both *cis* and *trans* curves. Binding affinity constants ($K_d$ values) were determined by fitting the data to a standard 1:1 binding equation (Eq. 1). The experiment was repeated, and arithmetic mean value for $K_d$ values of two independent measurements were taken (see Fig. S6).

**Thermal *cis*-to-*trans*-isomerisation**. For the determination of thermal relaxation kinetics, UV–Vis absorption spectra in the range of 200–600 nm were recorded with a Cary 100 Bio spectrophotometer (Agilent Technologies, Santa Clara, CA, USA) equipped with a $6 \times 6$ multicell block Peltier thermostat (Series II). Maximum conversion to the *cis* state was achieved by illuminating dark-adapted samples with a 370 nm laser until no further changes in the absorption spectrum were noted ($\approx$5 min). Subsequently, a series of spectra were recorded at adequate time points to cover the full kinetic trace at the respective temperature (2.5–20 h). The highest temperature chosen was at 40 °C, to ensure that the protein remains folded (see Fig. S4). The intensity change at the absorption maximum of the *trans*-state (370 nm) was fitted to a mono-exponential function in order to extract the time constants (see Fig. S7).

**Reporting summary**. Further information on research design is available in the Nature Research Reporting Summary linked to this article.

## Data availability
Data supporting the findings of this manuscript are available from the corresponding author upon reasonable request. A reporting summary for this article is available as a Supplementary Information file. Source data are provided with this paper.

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

## Acknowledgements

We thank Gerhard Stock and his coworkers for alerting us, based on preliminary MD simulations, that the first lysine side chain of a longer peptide we initially explored (KKETWV) is directed towards the $\alpha_3$-helix and may form a salt-bridges with it. We also thank Michal Shoshan for help with synthesis of the peptide, the Functional Genomics Center Zurich, especially Serge Chesnov and Birgit Roth, for their work on the mass spectrometry and amino-acid analysis, and Eva Freisinger and her group, in particular Jelena Habjanic, for discussions and help with protein expression. The work has been supported by the Swiss National Science Foundation (SNF) through the NCCR MUST and Grant 200020B_188694/1.

## Author contributions

O.B. designed research, took and analysed data, and contributed to the writing of the paper, B.J. took and analysed data and contributed to the writing of the paper, P.H. designed research, analysed data, and contributed to the writing of the paper.

## Competing interests

The authors declare no competing interests.
