## [Peer Review File · Nature Communications]

REVIEWER COMMENTS

Reviewer #1 (Remarks to the Author):

This is an intriguing paper that presents some remarkable data on a novel photoswitchable, allosteric protein. It is a great system, amenable to detailed study by both experimental and computational methods. Thus, I think the work is of wide general interest.

However, as it stands, I think the data is incomplete and the interpretation is presented as fact when I feel it is speculative. I think further data should be acquired/presented and the interpretation scaled back.

Data is presented in Figs 2 and 3.

- The delta CD signal does suggest a change in helix content, but is the intensity consistent with full unfolding of α_3 as depicted? It seems very small. What are the error bars on these measurements?
- Panel b presents titrations labeled trans and cis. This is at some %cis at a photostationary state? What %? Is analysis corrected for the presence of residual trans? What protein concentration was used? 10 μM ?
- Raw data for each titration should be given in the SI for each temperature to allow the reader to assess quality. Error bars should be provided (and error bars added to Fig.3). The data show a remarkable effect of photoisomerization at 40 degrees. And the reversal of the effect at lower temperatures is also remarkable. This alone is noteworthy.
- In panel C relaxation curves are presented (again time constants and error estimates should be given in SI, (and error bars added to Fig.3)). What peptide concentration was used for saturation? The behaviour is intriguing. However, relaxation data should be provided for each of the peptide concentrations used in the titrations to allow a landscape to be developed. What happens to the relaxation rate at low ligand concentrations at 40 degrees?

The second half of Section II then proceeds to a discussion of enthalpies of binding and of isomerization. The rationale for using focusing on enthalpies rather than of Gibbs free energies is given rather circuitously. Why not make the argument first to use focus on enthalpies? Even so, I am not really convinced that entropy is not playing a role in the isomerization event. Certainly, complex changes in dynamics are to be expected for this system based on previous NMR observations.

As the authors note, for the cis-state of the photoswitch, ligand binding is endothermic and entropy driven. This is quite unusual, but the explanation given is highly speculative. It seems this system is ripe for detailed analysis by simulation.

Lastly, I am not sure that the concept "introduced here" of an allosteric force is really useful. In fact, I find it misleading to think of mechanical force ("like a loaded spring") in this context. How does the calculated force generated compare to estimates from force spectroscopy measurements?

Reviewer #2 (Remarks to the Author):

The authors construct an allosteric protein from a PDZ3 domain by chemically attaching a photoswitchable azobenzene moiety to one helix. The isomerization state of this moiety affects the binding of a peptide at a distal site. The authors characterize the allosteric effect in both directions (the isomerization state on peptide affinity and peptide binding's effect on the kinetics of isomerization). The allosteric protein has some interesting thermodynamic properties. The authors use this characterization to introduce the concept of an allosteric force. Although the term, as defined here, is new to me, I do not know if it is new to the literature or if such a force has been calculated for other allosteric proteins. I am not aware of any paper, but it seems like a concept that would have been discussed before.

Here are my suggestions for improving the manuscript:

1. End of second paragraph of introduction: "true allostery" what is meant by "true"? Is there such a thing as false allostery?
2. Beginning of the second paragraph of the results: in what sense is the peptide non-proteogenic? Is it just meant to indicate that it is not a protein? That seems obvious by its length and that it is called a peptide.
3. Middle of second paragraph: "despite the fact" I don't understand why this phrase is used. The x-ray structure for a very similar peptide indicates one binding site. The expectation is that KETWV binding would also fit to one binding site. No, it's not a given that it will be one binding site, but the phrase "despite the fact" connotes that this is contrary to what is expected, and I disagree with that.
4. First paragraph of section 2. Even though the (log) values are technically in Fig 3a, it would help to have a table (supplement is fine) of the K_d values at 10, 21 and 40 degrees. I would also suggest stating early on in this paragraph that affinity in the trans state greatly increases with temperature, whereas affinity in the cis state slightly decreases with temperature. Yes, you can get this, with some thinking, from the logK_d vs. inverse temperature plots of Fig 3a, but a simple statement like that would have helped me quickly understand what the temperature effects were.
5. First paragraph of discussion "even larger than cleavage of the helix". I suggest the authors state the magnitude of the effect on affinity in this study so readers don't have to look it up to see how much larger the effect is.
6. Second paragraph of the discussion: "introduces an action-reaction principle of allostery, according to which allosteric communication is bidirectional". What is meant by "introduces." All allosteric communication is bidirectional (thermodynamic cycles tell us this). Although the specifics of this protein are new, the general phenomena the authors described is not new, so I am confused by what is meant by "introduces." The authors should note that many studies have observed allostery in both directions.
7. Fig 4a: Why are there no back arrows for the cis-to-trans isomerization? I think they should be there. Also, I assume the thicker arrow for the ligand-bound isomerization is meant to indicate a faster rate. But the thicker arrow, to me, appears shorter than the corresponding thinner one for the ligand unbound transition. But this is an optical illusion, I think, coming in part from the arrows being offset from other. This is truly a minor point, but I suggest aligning the arrows.
8. I find the introduction of the concept of an allosteric force interesting. But can the authors explain what new insight is gained by calculating such a force? What does the concept allow us to do that we couldn't before? Also, I worry that it will be misunderstood by those still clinging to the "pulley-and-lever" mechanical view of allostery and mislead those new to the concept of allostery down that road.
9. The authors take the energy difference for the cis to trans isomerization and divides it by the change in distance between the two anchor points of the azobenzene moiety to calculate the force. But isn't this an approximation that assumes the relative movement of two rigid masses at the two anchoring points? In reality, a variety of atoms are moving relative to each other in this transition. Maybe this approximation is what is meant by the authors saying "in essence" but I think it would be helpful if the authors more explicitly explained their assumptions/approximations in this calculation, given this paper is defining an apparently new term "allosteric force."
10. Can the authors expand on how allosteric force might be calculated and applied into other types of allosteric proteins? Is it limited to ligand binding driving a relatively simple conformational change?

Reviewer #3 (Remarks to the Author):

In this study, Hamm and coworkers advance their use of a photoactivatable conformational change in PDZ3 from PSD-95. They link the photoswitch to ligand binding in an allosteric arrangement, showing that allosteric control in this system is bidirectional. The photoswitch is an azobenzene moiety, and cis-trans modulation of this switch drastically affects ligand binding affinity at a distal site; moreover, peptide binding at that site reciprocally modulates the azobenzene cis-trans kinetics. What separates this from previous work is that the azobenzene moiety is attached to the 3rd helix in PDZ3 such that isomerization likely unfolds that helix, which is known to modulate peptide binding affinity. A major point of significance is that this establishes this PDZ3 (from PSD-95) system as arguably the smallest bona fide allosteric protein, with the protein only being ~100 amino acids and monomeric. This PDZ domain has previously been shown to behave in an allosteric manner, but this work technically proves the allostery by linking two discrete, dynamic processes (azobenzene isomerization and ligand binding). Interestingly, introduction of the azobenzene unit shows up to a 120-fold difference in peptide binding affinity between cis and trans forms of azobenzene-PDZ3, which is a much larger allosteric binding enhancement compared to the wild-type protein shown previously.

The quality of the work is overall high, with an elegant approach, and the manuscript is mostly well-written, though some details are left out. The major concern I have is with the equilibrium binding assays in Figure 2b, as illustrated by the following points:

1. The authors need to better explain "initial ligand concentration" in Figure 2b. PDZ was titrated into ligand, so how were the curves reconstructed? More detail needs to be given.
2. In the equilibrium binding experiments, is it assumed that when PDZ and ligand are mixed that the cis/trans state of the PDZ domain is preserved? If not, ascribing binding affinities for each would be subject to convolution errors. For example, binding of peptide might convert PDZ from trans to cis, depending on the temperature, and thus the curves generated for Figure 2b would not be from pure cis or trans states. One needs a binding model. With enough data, it should be possible to fit out the populations of cis and trans PDZ conformers. With the variable temperature data, it is probably possible to fit out the intrinsic equilibrium constant between cis and trans forms and therefore obtain true K_d values for cis and trans forms.
3. Related to the point above, the authors might look to the this 2015 JACS paper by Melacini and coworkers for obtaining state-specific binding affinities in systems undergoing a conformational equilibrium: JACS (2015), 137, 10777-10785.

Reviewer #1:

Reviewer General Comment: *This is an intriguing paper that presents some remarkable data on a novel photoswitchable, allosteric protein. It is a great system, amenable to detailed study by both experimental and computational methods. Thus, I think the work is of wide general interest. However, as it stands, I think the data is incomplete and the interpretation is presented as fact when I feel it is speculative. I think further data should be acquired/presented and the interpretation scaled back.*

Data is presented in Figs 2 and 3.

Reviewer Comment 1: *The delta CD signal does suggest a change in helix content, but is the intensity consistent with full unfolding of $\alpha 3$ as depicted? It seems very small. What are the error bars on these measurements?*

Author Response: We now quantified the effect by writing on page 2:

“The change in absolute ellipticity at 222 nm between the protein in trans and cis states is $\approx 5\%$, as compared to complete unfolding of the protein induced by urea (see SI, Fig. S5). Given that the $\alpha 3$ -helix accounts for $\approx 6\%$ of the overall protein, we conclude that photoswitching significantly perturbs the conformation of the distal $\alpha 3$ -helix.”

Details of the analysis are given in SI around Fig. S5.

Reviewer Comment 2: *Panel b presents titrations labeled trans and cis. This is at some %cis at a photostationary state? What %? Is analysis corrected for the presence of residual trans?*

Author Response: To clarify that point, we introduced on page 2:

“In all subsequent experiments, the trans-state has been measured as dark-adapted protein (for at least 24h), in which case it reaches essentially 100% of the trans-state. The cis-state has been measured during, or very shortly after, illumination at 370 nm. We verified by UV/VIS spectroscopy (see SI, Fig. S3) that the photo-equilibrium then reaches $>85\%$ cis. None of the reported numbers in the cis state have been corrected for the small residual trans-contribution.”

Reviewer Comment 3: *What protein concentration was used? 10 μM ?*

Author Response: The labelling of the x-axis of Fig. 2b was actually wrong (it has been the protein concentration that has been varied), and has been corrected. In addition, the concentration of the peptide ligand is now specified in the caption:

“Fraction of bound ligand as a function of initial protein concentration $[P_0]$ with the initial ligand concentration $[L_0]$ kept fixed at 15 μM .”

Reviewer Comment 4: *Raw data for each titration show be given in the SI for each temperature to allow the reader to assess quality. Error bars should be provided (and error bars added to Fig.3).*

Author Response: Error bars have been added to Fig. 3a, and added all titration curve and relaxation curves to SI (Fig. S6), as now indicated in the caption of Fig. 3:

“All titration and relaxation measurements behind these data points are given in SI as Figs. S6 and S7.”

Reviewer Comment 5: *The data show a remarkable effect of photoisomerization at 40 degrees. And the reversal of the effect at lower temperatures is also remarkable. This alone is noteworthy.*

Author Response: Agreed, and we underline this now by adding a sentence: “This inversion of behaviour in a relatively small temperature range is remarkable.” on page 3

Reviewer Comment 6: *In panel C relaxation curves are presented (again time constants and error estimates should be given in SI, (and error bars added to Fig.3)). What peptide concentration was used for saturation? The behaviour is intriguing.*

Author Response: Error bars have been added to Fig. 3b, and we added the information of saturation on page 3:

“(using a 40-times excess of peptide to ensure 100% binding)”

Reviewer Comment 7: *However, relaxation data should be provided for each of the peptide concentrations used in the titrations to allow a landscape to be developed. What happens to the relaxation rate at low ligand concentrations at 40 degrees?*

Author Response: We wouldn't expect any change at 40°C, as there is no measurable effect even for 100% bound vs free protein. Independent of that, exchange of ligand-binding and unbinding is expected to happen on a sub-second timescale. i.e., the timescale of k_{on} and k_{off} , which is much faster than thermal *cis-to-trans* isomerization (hours). Hence, for partial binding at other temperatures, we simply expect a rate that is the weighted average of the rates with ligand bound vs free protein. With all due respect, we don't think that this discussion would add anything to the paper, and we would prefer not to include it into the paper.

Reviewer Comment 8: *The second half of Section II then proceeds to a discussion of enthalpies of binding and of isomerization. The rationale for using focusing on enthalpies rather than of Gibbs free energies is given rather circuitously. Why not make the argument first to use focus on enthalpies?*

Author Response: We turned around the flow of arguments and first motivate why enthalpy is the property to consider in the paragraph starting with (page3): “Entropy plays a major role...”, and subsequently discuss the thermodynamic cycle in the paragraph starting with a new sentence:

“Given that the isomerization kinetics is mostly controlled by the activation enthalpy of thermal cis-to-trans isomerisation, we consider in Fig. 4a a thermodynamic cycle of enthalpy H.”

Reviewer Comment 9: *Even so, I am not really convinced that entropy is not playing a role in the isomerization event. Certainly, complex changes in dynamics are to be expected for this system based on previous NMR observations.*

Author Response: Entropy certainly plays a big role in the *response of the protein* to isomerization of the photo-switch, but for the rate-limiting limiting step of isomerization itself, apparently it does not. This is an experimental observation, as discussed in the context of the Eyring equation. To further rationalize this result in terms of a microscopic picture, we added the following sentence on page 3:

“In other words, the rate limiting step of barrier crossing affects the central N=N group of the azobenzene moiety only very locally, as evidenced by MD simulations of light-induced isomerization reactions in similar azobenzene-protein systems.^{26,40}”

Reviewer Comment 10: *As the authors note, for the cis-state of the photoswitch, ligand binding is endothermic and entropy driven. This is quite unusual, bit the explanation given is highly speculative. It seems this system is ripe of detailed analysis by simulation.*

Author Response: We agree that it is a speculative explanation, and we want to be honest about it by adding the following sentence on page 5:

“While this certainly is a speculative explanation, we consider it a working hypothesis to be tested by MD simulations, which naturally include the calculation of forces.”

also mentioning that atomistic MD simulation will be able to analyze the effect further.

Reviewer Comment 11: *Lastly, I not sure that the concept “introduced here” of an allosteric force is really useful. In fact, I find it misleading to think of mechanical force (“like a loaded spring”) in this context. How does the calculated force generated compare to estimates from force spectroscopy measurements?*

Author Response: Firstly, we deleted the term “like a loaded spring”, to remove a little bit of that flavor from the discussion. Furthermore, we put the estimated force in relation to force spectroscopy on page 4:

“The force is sizeable in comparison to what is typically observed in pulling experiments on proteins (on the order of 10-100 pN). However, when comparing these numbers, one must keep in mind that force spectroscopy typically works on significantly longer length scales (1-10 nm).⁴⁴”

We also refer to our answers to Comments 8-10 of Reviewer 2, which are related.

Reviewer #2:

Reviewer General Comment: *The authors construct an allosteric protein from a PDZ3 domain by chemically attaching a photoswitchable azobenzene moiety to one helix. The isomerization state of this moiety affects the binding of a peptide at a distal site. The authors characterize the allosteric effect in both directions (the isomerization state on peptide affinity and peptide binding's effect on the kinetics of isomerization). The allosteric protein has some interesting thermodynamic properties. The authors use this characterization to introduce the concept of an allosteric force. Although the term, as defined here, is new to me, I do not know if it is new to the literature or if such a force has been calculated for other allosteric proteins. I am not aware of any paper, but it seems like a concept that would have been discussed before.*

Author Response: To the best of our knowledge, the term “allosteric force” indeed has not been used before.

Here are my suggestions for improving the manuscript:

Reviewer Comment 1. *End of second paragraph of introduction: “true allostery” what is meant by “true”? Is there such a thing as false allostery?*

Author Response: We agree that this is not a proper characterization of the system and we just deleted the word “true”.

Reviewer Comment 2. *Beginning of the second paragraph of the results: in what sense is the peptide non-proteogenic? Is it just meant to indicate that it is not a protein? That seems obvious by its length and that it is called a peptide.*

Author Response: We also removed the word “non-proteogenic”

Reviewer Comment 3. *Middle of second paragraph: “despite the fact” I don't understand why this phrase is used. The x-ray structure for a very similar peptide indicates one binding site. The expectation is that KETWV binding would also fit to one binding site. No, it's not a given that it will be one binding site, but the phrase “despite the fact” connotes that this is contrary to what is expected, and I disagree with that.*

Author Response: Agreed. We rephrased the sentence (page 2): “..., which however is the X-ray structure of a slightly different peptide”

Reviewer Comment 4. *First paragraph of section 2. Even though the (log) values are technically in Fig 3a, it would help to have a table (supplement is fine) of the Kd values at 10, 21 and 40 degrees. I would also suggest stating early on in this paragraph that affinity in the trans state affinity greatly increases with temperature, whereas affinity in the cis state slightly decreases with temperature. Yes, you can get this, with some thinking, from the logKd vs. inverse temperature plots of Fig 3a, but a simple statement like that would have helped me quickly understand what the temperature effects were.*

Author Response: We added Table 1 with all the data, and also introduced a sentence along the lines of what the referee suggests on page 3:

“The binding affinity greatly increases with temperature in the cis-state, while that of the trans-state decreases slightly. As a consequence, the effect inverts in the considered temperature range, with a ca. 120-fold greater binding affinity of the cis-state at 40 °C vs a ca. 12-fold stronger binding affinity in the trans state at 10°C. This inversion of behaviour in a relatively small temperature range is remarkable.”

Reviewer Comment 5. *First paragraph of discussion “even larger than cleavage of the helix”. I suggest the authors state the magnitude of the effect on affinity in this study so readers don’t have to look it up to see how much larger the effect is.*

Author Response: Agreed, we now write (page 4):

“... even larger (i.e., 120-fold at 40 °C) than cleavage of the helix (21-fold)”

Reviewer Comment 6. *Second paragraph of the discussion: “introduces an action-reaction principle of allostery, according to which allosteric communication is bidirectional”. What is meant by “introduces.” All allosteric communication is bidirectional (thermodynamic cycles tell us this). Although the specifics of this protein are new, the general phenomena the authors described is not new, so I am confused by what is meant by “introduces.” The authors should note that many studies have observed allostery in both directions.*

Author Response: This comment, together with the next one, helped us tremendously to sharpen the way we think about the system. The novelty of the approach is now made very explicit by adding Fig. 4a (the “conventional” thermodynamic cycle) and contrasting it to a modified Fig. 4b (our system), together with a new paragraph on page 4:

“Fig. 4a shows the “conventional” thermodynamic cycle, which is commonly plotted in the context of the two-state model of allostery.⁴¹⁻⁴³ In this case, all reactions, ligand-binding and unbinding as well as the transition between the active (P*) and resting state (P) of the protein, are equilibrium reactions

with free energy differences in the order of $k_B T$. Hence, thermally driven transitions are possible in both directions (Fig. 4a). Bidirectional communication in allostery is the direct consequence of such a thermodynamic cycle. The important difference in the present system (Fig. 4b) is that the cis-to-trans reaction has a driving force much larger than $k_B T$ and the reaction is unidirectional. Light, however, introduces an additional knob of control, as indicated by the dashed arrows in Fig. 4b. This knob introduces an action-reaction principle, according to which “allosteric control” is bidirectional (to contrast it to bidirectional “allosteric communication”).”

Furthermore, we changed the title of Sec. 2 into

“Controlling Allostery in Both Direction”,

and the first sentence in Discussion and Conclusion:

“By constructing a photocontrollable PDZ3 domain, we can control allostery between binding and allosteric site in both directions.”

Reviewer Comment 7. *Fig 4a: Why are there no back arrows for the cis-to-trans isomerization? I think they should be there. Also, I assume the thicker arrow for the ligand-bound isomerization is meant to indicate a faster rate. But the thicker arrow, to me, appears shorter than the corresponding thinner one for the ligand unbound transition. But this is an optical illusion, I think, coming in part from the arrows being offset from other. This is truly a minor point, but I suggest aligning the arrows.*

Author Response: The response to the previous Comment answers why there are no back arrows. Yes, the thicker arrow for the ligand-bound isomerization is meant to indicate a faster rate. The arrows have been adjusted and aligned.

Reviewer Comment 8. *I find the introduction of the concept of an allosteric force interesting. But can the authors explain what new insight is gained by calculating such a force? What does the concept allow us to do that we couldn't before? Also, I worry that it will be misunderstood by those still clinging to the “pulley-and-lever” mechanical view of allostery and mislead those new to the concept of allostery down that road.*

Author Response: As already discussed in response to Comment 11 of Reviewer 1, we deleted the term “loaded spring” to remove some of the mechanical flavor from the picture. However, we cannot avoid thinking in terms of forces, given that the action of the photoswitch is to apply a force onto the protein backbone, as now discussed by a new paragraph on page 4:

“... between the two anchoring points $\Delta x \approx 3\text{\AA}$, as estimated from MD simulations.²³ It is important to stress that this is not a distance change induced by a rigid entity, as the azobenzene moiety of the photoswitch would probably be. Rather, there are linkers with six to a certain extent flexible single bonds on each side between the benzene rings and the protein backbone. We therefore think of the

distance change as the result of a "force" induced by the azobenzene moiety and transmitted to the protein backbone via these linkers."

Reviewer Comment 9. *The authors take the energy difference for the cis to trans isomerization and divides it by the change in distance between the two anchor points of the azobenzene moiety to calculate the force. But isn't this an approximation that assumes the relative movement of two rigid masses at the two anchoring points? In reality, a variety of atoms are moving relative to each other in this transition. Maybe this approximation is what is meant by the authors saying "in essence" but I think it would be helpful if the authors more explicitly explained their assumptions/approximations in this calculation, given this paper is defining an apparently new term "allosteric force."*

Author Response: The paragraph added in response to Comment 8 also addresses this point.

Reviewer Comment 10. *Can the authors expand on how allosteric force might be calculated and applied into other types of allosteric proteins? Is it limited to ligand binding driving a relatively simple conformational change?*

Author Response: Having only one example, obviously, we cannot comment at this point on how general that mechanism is. Nevertheless, we restricted a little bit which type of chemical reactions we have in mind (for example, the mechanism would presumably not work for electron transfer reactions) by writing on page 5:

"... to control chemical reactions that involve conformational changes."

Regarding calculations, forces are naturally calculated in MD simulations, and we added a sentence as an outlook (page 5):

"... to be tested by MD simulations, which naturally include the calculation of forces".

Regarding Comment 8-10, we also refer to our response to Comments 10-11 of Reviewer 1, which are related.

Reviewer #3:

Reviewer General Comment: *In this study, Hamm and coworkers advance their use of a photoactivatable conformational change in PDZ3 from PSD-95. They link the photoswitch to ligand binding in an allosteric arrangement, showing that allosteric control in this system is bidirectional. The photoswitch is an azobenzene moiety, and cis-trans modulation of this switch drastically affects ligand binding affinity at a distal site; moreover, peptide binding at that site reciprocally modulates the azobenzene cis-trans kinetics. What separates this from previous work is that the azobenzene moiety is attached to the 3rd helix in PDZ3 such that isomerization likely unfolds that helix, which is known to modulate peptide binding affinity. A major point of significance is that this establishes this PDZ3 (from PSD-95) system as arguably the smallest bona fide allosteric protein, with the protein only being ~100*

amino acids and monomeric. This PDZ domain has previously been shown to behave in an allosteric manner, but this work technically proves the allostery by linking two discrete, dynamic processes (azobenzene isomerization and ligand binding). Interestingly, introduction of the azobenzene unit shows up to a 120-fold difference in peptide binding affinity between *cis* and *trans* forms of azobenzene-PDZ3, which is a much larger allosteric binding enhancement compared to the wild-type protein shown previously.

The quality of the work is overall high, with an elegant approach, and the manuscript is mostly well-written, though some details are left out. The major concern I have is with the equilibrium binding assays in Figure 2b, as illustrated by the following points:

Reviewer Comment 1: The authors need to better explain “initial ligand concentration” in Figure 2b. PDZ was titrated into ligand, so how were the curves reconstructed? More detail needs to be given.

Author Response: The labelling of the x-axis of Fig. 2b was actually wrong (it has been the protein concentration that has been varied), and has been corrected. The “initial protein concentration”, as well as the way how the curves were constructed, are now explained with the help of Eq. 1, and the accompanying text (page2-3):

“The data were analyzed according to a two-state binding equilibrium:

$$K_D = \frac{[P] \cdot [L]}{[PL]} = \frac{([P_0] - [PL]) \cdot ([L_0] - [PL])}{[PL]}$$

where $[P]$, $[L]$, and $[PL]$ are the concentrations of free protein, free ligand and protein-ligand complex, respectively, after the system reached equilibrium, and $[P_0]$ and $[L_0]$ the initial (starting) concentrations of protein and ligand.”

Reviewer Comment 2. In the equilibrium binding experiments, is it assumed that when PDZ and ligand are mixed that the *cis/trans* state of the PDZ domain is preserved? If not, ascribing binding affinities for each would be subject to convolution errors. For example, binding of peptide might convert PDZ from *trans* to *cis*, depending on the temperature, and thus the curves generated for Figure 2b would not be from pure *cis* or *trans* states.

Author Response: This Comment is closely related to Comment 2 of Reviewer 1, and we refer to the answer given above. In brief, we can measure the amount of *cis* and *trans* via the UV/VIS spectrum during all experiments (see new Fig. SS3), and verify experimentally that binding of the peptide does not convert photoswitch from *trans* to *cis*”.

Reviewer Comment 3 One needs a binding model. With enough data, it should be possible to fit out the populations of *cis* and *trans* PDZ conformers. With the variable temperature data, it is probably

possible to fit out the intrinsic equilibrium constant between cis and trans forms and therefore obtain true Kd values for cis and trans forms.

Author Response: As explained in the context of Comments 6,7 of reviewer 2, the *cis-to-trans* reaction is not in a chemical equilibrium, see **new Figs. 4a vs 4b**, as well as their discussion on page 4.

Reviewer Comment 4. *Related to the point above, the authors might look to the this 2015 JACS paper by Melacini and coworkers for obtaining state-specific binding affinities in systems undergoing a conformational equilibrium: JACS (2015), 137, 10777-10785.*

Author Response: We cited the suggested paper in the context of the two-state model of allostery (**Ref. 43**), which shows a picture similar to Fig. 4a. Independent of that, as explained in the context of Comments 6,7 of reviewer 2, *cis* and *trans* states are not in a chemical equilibrium, hence it would not be meaningful to apply the methodology of that paper here.

REVIEWER COMMENTS

Reviewer #1 (Remarks to the Author):

This version of the manuscript is much clearer. The experimental data are clear and well reported, and remarkable. However, I find the interpretation and discussion problematic. Figure 4a shows a reaction scheme with 4 states and the conventional thermodynamic cycle would connect the Gibbs free energies of these states. Adding the curved arrows labeled "light" in Fig. 4b could imply a description of the system when the light is on, but then of course electronically excited states would have to be included and the whole picture is more complicated. The light is used here as a means of perturbing the system to put it transiently out of equilibrium. The system with the light off is really that in Fig. 4a, except that cis/trans equilibria are very much on the side of trans. If the equilibrium ratios of cis and trans states of the system could be measured they should be perturbed by ligand binding according to the thermodynamic cycle. Since the % cis at equilibrium is extremely small however, this would be very hard to do.

What is observed here is an effect of ligand binding on the rate of cis to trans isomerization. The rate is determined by the barrier height as the authors note. It seems to me that thermodynamics of the system place no constraints on the relative heights of any of the barriers connecting the 4 states. That is, one might see an effect of ligand binding on the thermal reversion rate or one might not, and the system could still be coupled allosterically. This is observed with other (natural) photoswitchable proteins. For example Lungu et al saw no effect of ligand binding on the thermal reversion rate of a LOV domain (<https://www.ncbi.nlm.nih.gov/pmc/articles/PMC3334866/>); whereas Morgan et al saw a large effect with a PYP domain: <https://pubmed.ncbi.nlm.nih.gov/20363227/>. Early work by Shinkai on azobenzene based chelators also found clear effects of ligand binding on rates of isomerization (in a much simpler system)(<https://pubs.acs.org/doi/10.1021/ja00391a021>). The observation that ligand binding does affect isomerization here clearly means the barrier height is affected by ligand (and probably the enthalpy more so than the entropy as the authors suggest), but how this is connected to the enthalpy of ligand binding is probably not simple. I don't think the claim that the barrier must be 92% cis-like is warranted for such a complex system.

Reviewer #2 (Remarks to the Author):

My comments have been satisfactory addressed.

Reviewer #1

We would like to thank the Reviewer for his/her deep thinking about the manuscript, and the critical and constructive comments, which we address below:

Reviewer Comment 1: *This version of the manuscript is much clearer. The experimental data are clear and well reported, and remarkable. However, I find the interpretation and discussion problematic. Figure 4a shows a reaction scheme with 4 states and the conventional thermodynamic cycle would connect the Gibbs free energies of these states. Adding the curved arrows labeled “light” in Fig. 4b could imply a description of the system when the light is on, but then of course electronically excited states would have to be included and the whole picture is more complicated. The light is used here as a means of perturbing the system to put it transiently out of equilibrium. The system with the light off is really that in Fig. 4a, except that cis/trans equilibria are very much on the side of trans. If the equilibrium ratios of cis and trans states of the system could be measured they should be perturbed by ligand binding according to the thermodynamic cycle. Since the % cis at equilibrium is extremely small however, this would be very hard to do.*

Author Response: We would like to start with mentioning that we introduced the comparison of Fig. 4a vs Fig. 4b in response to Reviewer 2 of the previous round, who was satisfied with that change, but now seems to make Reviewer 1 unhappy. We tried to find a solution in between. That is, we updated Fig. 4b, emphasizing that indeed the *cis-trans* isomerization is still an equilibrium reaction with the equilibrium very much on the *trans* side (indicated by a new, short *trans-to-cis* arrows), and that light is used for a transient perturbation (indicated by the flash and the curved arrow). We like the phrase of the Reviewer: “*put transiently out of equilibrium*” and took it over. The new features of Fig. 4b are mentioned in the figure caption and in the text on page 4:

“... (indicated by the short, *trans-to-cis* arrows in Fig. 4b)”

and

“.. that transiently puts the system out of equilibrium, as indicated by the flash and the curved dotted arrows in Fig. 4b.”

In addition, we slightly changed Fig. 4c in such a way that the difference in barrier heights is visually more obvious.

Reviewer Comment 2: *What is observed here is an effect of ligand binding on the rate of cis to trans isomerization. The rate is determined by the barrier height as the authors note. It seems to me that thermodynamics of the system place no constraints on the relative heights of any of the barriers connecting the 4 states. That is, one might see an effect of ligand binding on the thermal reversion rate or one might not, and the system could still be coupled allosterically. This is observed with other (natural) photoswitchable proteins. For example Lungu et al saw no effect of ligand binding on the*

thermal reversion rate of a LOV domain (<https://www.ncbi.nlm.nih.gov/pmc/articles/PMC3334866/>); whereas Morgan et al saw a large effect with a PYP domain: <https://pubmed.ncbi.nlm.nih.gov/20363227/>. Early work by Shinkai on azobenzene based chelators also found clear effects of ligand binding on rates of isomerization (in a much simpler system) (<https://pubs.acs.org/doi/10.1021/ja00391a021>). The observation that ligand binding does affect isomerization here clearly means the barrier height is affected by ligand (and probably the enthalpy more so than the entropy as the authors suggest), but how this is connected to the enthalpy of ligand binding is probably not simple. I don't think the claim that the barrier must be 92% cis-like is warranted for such a complex system.

Author Response: Strictly speaking, the Reviewer is right, of course, and we said that already in the original version of the manuscript: "The driving force *per se* does not necessarily determine the rate of a chemical reaction; for that the activation energy is relevant." Nevertheless, very often a larger driving force of a chemical reaction does imply a faster reaction rate, known as "linear free energy relationship" in chemical kinetics. There are counter examples, the most prominent probably being the inverted regime in Marcus theory of electron transfer. Marcus predicted the inverted regime in the mid 50's of the last century, but it took 30 years until a molecular system was found that actually showed that behavior. Only then he received the Nobel prize, emphasizing how wide-spread the observation has been (and still is) that a larger driving force does in fact imply a faster reaction rate in the majority of cases.

The situation that is probably the closest to the one discussed here is the " Φ -value analysis", which applies the linear free energy relationship to the protein-folding problem. We would argue that protein-folding is equally complex as the problem discussed here (maybe even more complex, in the sense that a protein completely changes its structure), yet it has been shown that the Φ -value analysis is useful to characterize degree of native structure in the transition state around a certain amino acid. Characterizing the transition state is the aim also here. To address the comment of the Reviewer, we added a discussion on page 4:

"Presumably, the barrier is mostly *cis*-like since the event of *cis*-to-*trans* isomerisation is the initial process, which is fast, while the protein adapts to the change on a much slower timescale.

The spirit of this analysis is the same as that of a " Φ -value analysis" used in the context of protein folding,⁴⁷ which also relates the folding free energy, determined thermodynamically, to the height of the folding barrier, measured kinetically. The Φ -value analysis is closely related to the "linear free energy relationship" common in chemical kinetics, which is used as a means of analyzing structures of transition states, and acknowledges that the activation energy of a chemical reaction is often correlated with its driving force. The linear free-energy relationship has been discussed in the context of allostery before.⁴⁸ Just like for a Φ -value analysis, there will be cases where ligand binding has little effect on the isomerisation kinetics of a chromophore in a protein,³⁹ and others with a large effect.³⁸"

where we also cite the papers mentioned by the Reviewer. We first introduce these papers on page 3:

“... there are a few reports in literature with similar observations,^{37–39} which however have been discussed in a different context.”

Furthermore, when introducing Fig. 4c we recapture where the numbers come from (page 4):

“..., where ΔH_{cis} and ΔH_{trans} have been determined from the binding equilibria (Fig. 3a), and the difference in barrier heights $\Delta\Delta H^\ddagger = \Delta H^\ddagger_{\text{PL}} - \Delta H^\ddagger_{\text{P}}$ from the thermal *cis*-to-*trans* isomerisation rates (Fig. 3b)”

Finally, we changed the labelling of the x-axis of Fig. 4c from “Isomerisation Coordinate” into “Reaction Coordinate”, since it also includes the protein response.

The additions implied a change in the sequence of the various paragraphs in “Discussion and Conclusion”, for a straighter line of arguments.

Reviewer #2

Reviewer Comment: *My comments have been satisfactory addressed.*

Author response: No action needed.

REVIEWERS' COMMENTS

Reviewer #1 (Remarks to the Author):

My comments have been satisfactory addressed.

Reviewer #1

Reviewer Comment: *My comments have been satisfactory addressed.*

Author Response: No action is needed. We want to thank the Reviewer again for his/her great contribution to the paper.